# Investigating the multi-millennial evolution and stability of the Greenland ice sheet using remapped surface mass balance forcing

Charlotte Rahlves<sup>1,2</sup>, Heiko Goelzer<sup>1</sup>, Andreas Born<sup>2</sup>, and Petra M. Langebroek<sup>1</sup>

<sup>1</sup>NORCE Research AS, Bjerknes Centre for Climate Research, Bergen, Norway

<sup>2</sup>Department of Earth Science, University of Bergen, Bjerknes Centre for Climate Research, Bergen, Norway

**Correspondence:** Charlotte Rahlves (chra@norceresearch.no)

Abstract. Surface mass balance (SMB) forcing for projections of the future evolution of the Greenland ice sheet with standalone modeling approaches has been commonly derived from regional climate models (RCMs) on a fixed ice sheet topography. However, over long time scales, changes in ice sheet geometry become substantial, and using SMB fields that do not account for these changes can introduce non-physical biases. Therefore, conducting projections for the long term evolution and stability of the Greenland ice sheet usually requires a computationally expensive coupled climate-ice sheet modeling setup. In this study we use a SMB remapping procedure to capture the first order feedbacks of the coupled climate-ice sheet system within a computationally efficient stand-alone modeling approach. Following a remapping procedure that was originally developed to apply SMB forcing to a range of modeled steady-state ice sheet geometries, we produce SMB forcing that adapts to the changing ice sheet geometry as it evolves over time. SMB fields from a regional climate model are translated from a function of absolute geographic location to a function of surface elevation, allowing for SMB updates when elevation changes. To reflect the heterogeneous elevation response across the ice sheet we separate the ice sheet into 25 regional drainage basins, which allows for a spatially resolved adjustment of SMB. We evaluate this approach using forcing from multiple emission scenarios from the CMIP6 archive and compare the results with those from standard parameterizations of the SMB—elevation feedback. Our results show that the remapping method better preserves the structure of the ablation zone and reduces non-physical biases compared to conventional SMB—elevation feedback parameterizations, while still leveraging high-quality forcing data.

#### 1 Introduction

The future evolution and stability of the Greenland ice sheet remain critical topics in climate research due to their implications for global sea-level rise. Although projections indicate only limited mass loss of the ice sheet within this century (Goelzer et al., 2020a; Hofer et al., 2020; Payne et al., 2021; Rahlves et al., 2025), long-term simulations suggest that sustained elevated temperatures could lead to substantial mass loss or even a complete decline of the ice sheet over the next millennia (Ridley et al., 2009; Robinson et al., 2012; Gregory et al., 2020; Bochow et al., 2023; Petrini et al., 2025).

One of the primary drivers of ice sheet evolution is surface mass balance (SMB), which represents the net gain or loss of mass at the surface of the ice sheet. SMB integrates key processes such as snowfall, melt, refreezing, and runoff, and is highly sensitive to atmospheric conditions. A critical mechanism influencing SMB over time is the SMB-elevation feedback. As the

ice sheet loses mass, its surface elevation lowers, exposing the ice sheet to warmer temperatures at lower altitudes due to the adiabatic lapse rate. This process further enhances surface melt, reinforcing mass loss and altering the SMB. Edwards et al. (2014a) demonstrated that this feedback is positive across the entire ice sheet in projections until 2200, amplifying mass loss and sea-level contributions. Failing to account for this feedback in long-term simulations can therefore lead to significant biases, likely underestimating future mass loss.

30

55

To accurately capture the interplay between evolving ice topography and atmospheric conditions in an ice sheet simulation, one would ideally use a two-way coupled climate-ice sheet system. Fully coupled Earth System Model-Ice Sheet Model (ESM-ISM) simulations dynamically integrate atmosphere, ocean, land, and ice sheet components, capturing large-scale climate feedbacks. This approach has been used in several studies investigating the future evolution of the Greenland ice sheet (Muntjewerf et al., 2020; Smith et al., 2021; Goelzer et al., 2025b; Haubner et al., 2025). However, ESMs are limited by coarse spatial resolution and high computational demands, which reduce their accuracy in representing regional-scale processes. Simplifications in process representation are often necessary to make models computationally feasible over multi-millennial timescales.

Earth System Models of Intermediate Complexity (EMICs) provide a computationally efficient alternative for simulating long-term ice sheet—climate interactions. While EMICs include essential Earth system processes and can be coupled with ice sheet models for long-term simulations, they operate at relatively low spatial resolution and/or rely on simplified physics for atmosphere, ocean and land-surface processes (Ridley et al., 2009; Gregory et al., 2020). As a result of their low spatial resolution, they fail to resolve the critical SMB features near the ice sheet margin. Although it is possible to correct these biases, doing so relies on high-resolution data (Hoang et al., 2025).

Recent modeling frameworks apply elevation class downscaling, which enables stand-alone simulations that incorporate SMB-elevation feedbacks based on ESM output (Sellevold et al., 2019; Petrini et al., 2025). In these cases, the ice sheet geometry evolves in the ISM, but the atmosphere and land surface are not updated accordingly. This one-way coupling can introduce biases in SMB and ice loss, and as shown by Feenstra et al. (2025), such setups differ from two-way coupled ESM ISM frameworks where feedbacks between evolving ice sheet geometry and climate forcing are explicitly represented. While elevation class approaches offer a computationally efficient way to account for elevation-dependent climate forcing and are particularly effective in addressing resolution mismatches between climate and ice sheet models, their main limitation lies in capturing lateral shifts in climate forcing, such as the inland migration of precipitation, temperature, and melt patterns that accompany ice sheet margin retreat. Because elevation class methods operate vertically within static grid columns, they cannot adequately represent the spatial displacement of SMB fields driven by evolving ice sheet geometry and retreating slopes. Furthermore, their tight integration within a coupled ESM often restricts their use as a flexible, stand-alone SMB module.

Regional Climate Model–Ice Sheet Model (RCM-ISM) coupling offers higher spatial resolution and a more detailed representation of atmospheric processes (e.g. Le clec'h et al., 2019). For example, in a coupled simulation using a high emission scenario extending to the year 2200, Delhasse et al. (2024) demonstrated that geometric changes of the ice sheet lead to a modification of local wind regimes, which affects SMB at the ice sheet margins. However, RCM-ISM simulations are computationally intensive restricting their feasibility for long-term projections. Furthermore, due to these computational constraints,

such simulations often explore only a limited number of forcings or scenarios, limiting their ability to capture the full range of possible ice sheet responses. In addition, RCMs require boundary conditions from ESMs, which introduces another layer of dependency and limits their use for long-term stand-alone simulations.

A more simplified but efficient approach is using a Positive Degree Day (PDD) model in a stand-alone ice sheet modeling setup. PDD models parameterize SMB based on temperature and precipitation, by estimating melt as a function of cumulated temperatures above freezing (e.g. Braithwaite, 1995; van de Wal, 1996; Wilton et al., 2016). These models typically use a linear temperature lapse rate to approximate elevation effects. While these models use a simplified representation of SMB processes, they are commonly used for studying the long-term evolution and stability of the ice sheet due to their computational efficiency (Huybrechts and de Wolde, 1999; Fürst et al., 2015; Aschwanden et al., 2019; Bochow et al., 2023).

The diurnal Energy Balance Model (dEBM) offers a physically-based alternative to empirical surface mass balance schemes (Krebs-Kanzow et al., 2021). It explicitly resolves the energy balance at the surface of the ice sheet. Unlike temperature-index or PDD models, dEBM accounts for changes in Earth's orbital parameters and atmospheric composition, which makes it particularly well-suited for paleoclimate and long-term future simulations where radiative forcing varies significantly over time.

RCM-forced stand-alone modeling approaches use SMB fields derived from RCMs, which are typically based on a fixed ice sheet topography. This method captures local processes with higher accuracy and is widely used for simulations on decadal to multi-centennial time scales (Edwards et al., 2014b; Goelzer et al., 2020b; Rahlves et al., 2025). However, over long time scales, significant changes in ice sheet geometry make it problematic to apply fixed-topography SMB fields, especially under high-emission scenarios. In such cases, retreat and thinning can substantially alter the SMB distribution. To account for the SMB-elevation feedback in stand-alone modeling, one common method is to adjust RCM-derived SMB fields based on local runoff gradients in response to elevation changes (Franco et al., 2012). Other approaches include lapse-rate corrections, which adjust SMB using temperature-based lapse rates, and SMB-gradient methods, which update SMB with locally derived elevation—SMB gradients (Helsen et al., 2012; Edwards et al., 2014a, b). While these methods are effective on centennial timescales, they may not adequately capture feedbacks that occur under substantial geometric changes in long-term, high-forcing scenarios.

To address these challenges, we introduce an SMB-remapping scheme that captures the first-order feedbacks of a coupled climate—ice-sheet system while retaining the computational efficiency of a stand-alone ice-sheet model. This method builds on the SMB anomaly (aSMB) remapping approach, originally developed by Goelzer et al. (2020b), which was designed to apply SMB forcing consistently across Greenland ice sheet models with varying initial geometries. In the original approach, anomalies relative to a reference climate were applied to a set of ice sheet topographies, enabling consistent/uniform application of SMB forcing to an ensemble of ice sheet models.

We extend this framework in two key ways. First, we enable a dynamic adaptation of the SMB forcing to the evolving geometry of the ice sheet, which allows the applied SMB fields to adapt to changes in elevation and ice extent over time. Second, we implement both aSMB remapping and total SMB remapping. The aSMB method retains the spatial structure of a reference SMB field and overlays evolving anomalies, while the total SMB remapping approach directly adjusts the full SMB field to reflect topographic changes. This distinction allows us to evaluate the sensitivity of long-term ice sheet evolution to the

choice of remapping method and to assess whether remapping total SMB values offers a more realistic ice sheet response to the applied forcing scenario.

Although the remapping approach does not explicitly resolve atmospheric feedbacks, such as changes in wind regimes, cloud cover, or precipitation patterns, it offers a computationally efficient and physically motivated alternative to fully coupled climate—ice sheet modeling, based on high-quality RCM output. By dynamically adjusting SMB fields in response to evolving ice sheet geometry, the method maintains a realistic structure of the ablation zone, which is critical for projecting long-term ice sheet retreat. While it operates as a one-way feedback, the similarity-based remapping structure may implicitly capture some regionally evolving processes, such as the inland migration of katabatic wind effects associated with a retreating ice margin.

In this study, we compare the results of our remapping approach with a commonly used SMB-elevation feedback parameterization, and assess their respective impacts on ice sheet dynamics and stability over multi-millennial timescales. We apply SMB remapping to investigate the long-term evolution and possible stability of the Greenland ice sheet under various emission scenarios from the Coupled Model Intercomparison Project Phase 6 (CMIP6) archive (Eyring et al., 2016).

The following section (Sect. 2) describes the ice sheet model, the experimental set-up, and the remapping method. Results are presented in Sect.3. We examine our simulations on a centennial time-scale in Sect.3.1 and on a millennial time-scale in Sect.3.2. Sect. 4 offers a discussion of the results, before we conclude with a summary in Sect. 5.

# 110 2 Experimental setup





We simulate the long-term evolution of the Greenland ice sheet under prescribed climate forcing scenarios using the Community Ice Sheet Model (CISM) (Lipscomb et al., 2019). This section describes the model configuration, initialization strategy, and the different approaches applied to SMB forcing in future climate projections.

The overarching goal of these experiments is to present the SMB remapping forcing approach, compare it to an established method that uses RCM-output alongside a parameterization for SMB-elevation feedback, and demonstrate its suitability for investigating the long-term response of the Greenland ice sheet to sustained climate forcing. We first evaluate the influence of different SMB-forcing strategies and associated feedbacks on modeled ice sheet evolution using dynamically downscaled forcing from NorESM2-MM under Shared Socio-economic Pathway (SSP) 5-8.5. To this end, we extent the forcing beyond 2100 and perform sensitivity experiments employing alternative SMB representations. We then apply the SMB-remapping scheme to additional ESM-scenario combinations from the CMIP6 archive, thereby exploring a wider range of future climates. For readability, MAR-derived SMB fields and the corresponding CISM simulations are referred to simply by the name of the forcing ESM and scenario. Note that when an experiment is labeled, for example, "NorESM2-MM-SSP5-8.5," this refers to the full three-part simulation chain: the ESM climate output used to force MAR at its boundaries, the SMB computed by MAR, and the SMB subsequently applied as forcing in CISM.

#### 125 2.1 Ice sheet model setup






CISM solves the depth-integrated Stokes equations using a vertically-integrated viscosity approximation (Goldberg, 2011). Simulations are run on a 4 km Cartesian grid; the ice column is discretized into 11 vertical layers that are progressively thinned toward the bed to resolve regions of high shear. Basal sliding follows a Weertman-type power law (Weertman, 1957). Any ice that becomes buoyant is removed immediately from the ice sheet domain, representing mass loss through iceberg discharge. Meltwater produced at the bed is likewise removed instantaneously.

## 2.2 Spin-up and Historical initialization

The initialization and spin-up protocol follows Rahlves et al. (2025). Bedrock topography and present-day ice sheet geometry are taken from BedMachine v3 (Morlighem et al., 2017). Bedrock topography is smoothed with a Gaussian filter for numerical stability before interpolation onto the 4 km model grid. The model undergoes a 5000-year spin-up, applying the annual mean surface mass balance (SMB) and surface temperature (ST) from ERA5 reanalysis (Hersbach et al., 2020) for 1960-1989, a period during which the Greenland ice sheet is assumed to have been close to climatic equilibrium (Van den Broeke et al., 2009). Vertical ice temperatures are initialised from an advection-diffusion balance between the prescribed ST and a geothermal heat flux following Shapiro and Ritzwoller (2004). During spin-up, basal friction parameters are nudged following Pollard and DeConto (2012), so that modelled surface elevations converge toward observations. This minimizes the influence of basal temperature on sliding and compensates for modeling uncertainties such as basal heat flux and bed roughness effects (Berends et al., 2023). The calibrated parameters remain fixed for the follow-up simulations. To reduce residual drift, the ice sheet undergoes an additional 1000-year relaxation on the adjusted bed friction field. The final ice geometry is defined as the initial state for the historical simulation, corresponding to the beginning of 1960. We subsequently perform a historical simulation over the 1960–2014 period. The historical run is forced with SMB and ST fields computed by MAR, which is driven at its boundaries by ERA5 meteorological variables. Outlet-glacier retreat is prescribed using a retreat mask (Slater et al., 2019, 2020) that best match observed changes (see Fig. 6 in Rahlves et al. (2025)). In this way, the response of the ice sheet to historical forcing is accounted for and carried forward into the projections. For details, see (Rahlves et al., 2025).

#### 2.3 Future climate and SMB forcing approaches

Future SMB forcing is taken from the NorESM2-MM SSP5-8.5 projection, dynamically downscaled over Greenland with MAR v3.12 (Fettweis et al., 2017; Lambin et al., 2022). As a first preprocessing step, the MAR SMB fields are conservatively interpolated from their native grid onto the 4 km CISM grid. We select NorESM2-MM as an illustrative CMIP6 driver. Although its 21st-century warming sits toward the lower end of the SSP5-8.5 spread, it still provides a strong enough perturbation to test long-term ice-sheet stability. Additional ESM–scenario combinations that span the broader CMIP6 range are explored later in Sect. 3.3. To keep the comparison focused on surface processes, ice-temperature evolution is switched off and only SMB forcing is varied. Oceanic forcing is not simulated explicitly; instead, its influence on outlet glaciers is represented via prescribed retreat masks. These masks are applied up to 2100 and then held fixed, such that glaciers cannot re-advance beyond

the mask positions after 2100, although further retreat within the mask remains possible.

Beyond 2100, the forcing is extended by averaging the final 20 years (2081–2100) and repeating this mean SMB value at annual time steps. We tested an alternative approach where the individual years within this 20-year window were randomly shuffled and repeated, and found that this does not significantly affect the results. No systematic drift is introduced, and the effect of shuffling is limited to minor, random variability. Throughout the experiment, the maximum difference in simulated mass loss remains below 45 Gt. We employ four forcing strategies to simulate ice sheet evolution over 10,000 years: SMB anomalies produced at fixed surface elevation, SMB anomalies at fixed surface elevation with parameterized SMB-elevation feedback, SMB anomaly remapping and remapping of total SMB.

For selected simulations we allow for a response of the Earth's crust to changes in ice load. Glacial-isostatic adjustment is simulated using an Elastic Lithosphere, Relaxing Asthenosphere (ELRA) model (Rutt et al., 2009), which approximates bedrock response to ice loading via a linear relaxation toward local isostatic equilibrium. We use a characteristic relaxation timescale of 3000 years.

#### 2.3.1 SMB anomalies produced at fixed surface elevation



As a baseline experiment, we compute the SMB at timestep t as the sum of a reference SMB and an ESM-derived anomaly:

$$SMB(t) = SMB\_ref + SMB\_anomaly(t)$$
 (1)

where SMB\_ref is the reference SMB (here, the ERA5-forced MAR mean over 1960–1989), and SMB\_anomaly(t) is the ESM anomaly at time step t, defined as:

$$SMB_{anomaly}(t) = SMB_{ESM}(t) - SMB_{ref}_{ESM}$$
 (2)

with SMB ref ESM denoting the mean ESM SMB over the same reference period.

Anomaly-based SMB methods such as this are widely used to address the potentially large mean-state biases in ESMs. By focusing on anomalies relative to a reference state, this method reduces the influence of absolute model biases and instead emphasizes the physical response of SMB to climate forcing (e.g., Goelzer et al. 2013; Ađalgeirsdóttir et al. 2014; Nowicki et al. 2020).

80 In the baseline experiments, we neglect any surface elevation changes when calculating the SMB forcing and directly apply MAR forcing (which is produced on a fixed ice sheet geometry) to the ice sheet model, meaning that the SMB-elevation feedback is not accounted for.

# 2.3.2 SMB anomalies at fixed surface elevation with parameterized SMB-elevation feedback

To incorporate SMB-elevation feedback, we apply a parameterization based on local vertical runoff gradients, following the ISMIP6 standard approach (Nowicki et al., 2020). The applied SMB in each grid cell is corrected for elevation changes using:

$$SMB\_applied = SMB(t) + \Delta h \frac{dRU}{dz}$$
(3)

where SMB(t) is the MAR SMB, produced on the fixed ice sheet geometry (see Eq. (1)),  $\Delta h$  is the elevation change relative to the reference elevation (of the initial ice sheet state) and  $\frac{dRU}{dz}$  is the local runoff gradient calculated from surrounding cells following Franco et al. (2012). We use runoff gradients instead of SMB gradients since precipitation variability introduces inconsistencies in SMB-elevation relationships. For details, see Sect. 6.1 and Fig. S11 in Franco et al. (2012).

This parameterization has limitations, particularly in regions with weak runoff gradients such as the interior of the ice sheet. Consequently, it is expected to perform poorly in scenarios with substantial ice sheet retreat and margin recession into the Greenland ice sheet interior.

## 2.3.3 Total SMB and SMB anomaly remapping

190

205

In their original study, Goelzer et al. (2020b) remapped SMB anomalies to a range of static ice-sheet geometries and outlined how the same idea could be applied to an evolving surface. We follow through on that idea by implementing the online formulation directly inside our ice-sheet model. In addition, we extend the framework to remap total SMB as well, so the model can be driven either by absolute SMB or by anomalies. Both options work in the same way, but with one key difference: because anomalies are defined relative to a reference SMB, aSMB remapping needs an extra elevation-correction term that remaps the vertical SMB gradient to the reference geometry, whereas full-SMB remapping already preserves the vertical structure (and hence the elevation feedback) by construction. In the following, we first outline the principle remapping method and then describe the additional term required for anomaly forcing in Sect. 2.3.4.

The remapping method reconstructs SMB as a function of elevation, thus preserving the effect of geometry-dependent changes in SMB when ice sheet margins retreat (advance). Instead of directly applying annual SMB or SMB anomaly fields from MAR to the ice sheet model, we transform the SMB product into an elevation-dependent function rather than a fixed spatial field (Fig. 1).

SMB values are stored in a lookup table, where SMB is stored as a function of surface elevation and regional drainage basin. This accounts for regional variations in the elevation-SMB relationship. These can be significant, particularly in areas with contrasting climatic conditions, such as wet regions with high accumulation rates and drier regions where SMB is primarily controlled by sublimation and low precipitation. To capture these regional differences, remapping is applied separately within 25 drainage basins following Mouginot et al. (2019). The resulting lookup table is then provided to the ice sheet model, allowing for dynamic SMB adjustments in response to evolving geometry, effectively stretching or shrinking the forcing to match the evolved ice sheet extent and elevation.

At each model time step, updated SMB values are obtained by interpolating within the lookup table based on the current local surface elevation and associated basin ID. This ensures that SMB forcing remains physically consistent with the evolving ice sheet topography throughout the simulation. The overall remapping procedure consists of two key steps:

#### **Step 1: Construction of an SMB Lookup Table:**

The original SMB forcing field is transformed into a lookup table where SMB values are stored as a function of surface elevation and regional drainage basin, rather than absolute geographic location. This involves the following steps:

- 1. Divide the ice sheet into 25 regional ice flow basins, based on observed hydrological divides (Fig. 2).
- 2. Within each basin, define elevation bands with 100 m intervals.
- 3. For each forcing time step (in our case each year) and for each elevation band:
  - (a) Identify all grid points within that elevation range.





- (b) Compute the median SMB across these grid points to smooth out local variations.
- (c) Store results in a lookup table: SMB = f(basin, elevation)

The elevation band width of 100 m represents a balance between capturing spatial variability in the SMB field and ensuring smoothness of the elevation–SMB relationship. We adopt the 100 m step size following Goelzer et al. (2020b), who selected it based on initial testing with a 15 km SMB product. Finer intervals may overfit noise or lead to instability, whereas coarser bands may smooth out key ablation-zone gradients. Because 100 m remains the best compromise between detail and stability, we adopt it unchanged here.

The lookup table is constructed prior to the ice sheet simulation and is, in principle, based on the ice sheet topography used to generate the SMB, which in this case is the MAR topography. However, in this study, we instead use the initial ice sheet geometry provided by the ice sheet model. This choice ensures internal consistency within the modeling framework and avoids potential interpolation artifacts or mismatches between the MAR topography and the model's initial state. Notably, the two geometries are very similar, so this substitution introduces minimal error while simplifying the setup. To improve stability in regions with sparse data, adjustments are made at both low and high elevations. At 0 m elevation, the SMB value from 100 m elevation is used instead of the sparsely populated 0–50 m range, ensuring a more reliable interpolation. Similarly, at high elevations, where SMB data is limited, the highest available SMB value is extrapolated up to 3500 m to maintain consistency and avoid discontinuities in the forcing data.

Figure 3 shows the lookup table generated for NorESM2-MM SSP5-8.5 forcing from 2015 to 2100. As expected, SMB declines both with decreasing elevation and over time as warming intensifies. Clear inter-basin variations are also visible, reflecting regional differences in accumulation and melt patterns.

As for the other approaches, forcing beyond 2100 is extended by averaging over the final 20 years (2081–2100). Thus, beyond 2100 the lookup table is held constant at this 2081–2100 mean.

#### Step 2: Remapping SMB to the Evolving Ice Sheet Geometry:

Once the lookup table is constructed, SMB can be dynamically remapped to the evolving ice sheet geometry during the ice sheet simulation at every model time step. This process involves:

- 1. Updating the ice sheet's surface elevation at each model time step.
- 2. For each grid cell, the updated local elevation and basin membership are used to obtain the SMB value from the lookup table. Because the table is discretized by elevation, the value is determined by linearly interpolating between the two bracketing elevation classes of the corresponding basin.

3. Along drainage basin boundaries, where cells may be influenced by more than one basin, SMB values are smoothed across adjacent basins.




When applying the remapped SMB to the ice sheet model grid, SMB values are thus reconstructed at each grid point from a combination of lookup tables from both the local and nearby basins. Within each basin, the SMB value at a given grid point is obtained by interpolating between the two bracketing elevation classes according to the cell's updated elevation. To ensure a smooth transition across basin boundaries, SMB values from surrounding basins are incorporated into the interpolation process. A proximity-based weighting scheme is applied, reducing the influence of neighbouring basins with increasing distance from the basin divide (Fig. 2). For example, at a basin divide with only one neighboring basin, the weighting gradually shifts from an equal contribution of both basins at the divide to a full contribution from the local basin at the basin center. This prevents abrupt changes and ensures spatial continuity in the SMB field. The exact calculation of the weighting is described in Sect. 2.2 of Goelzer et al. (2020b).

**Figure 1.** Schematic illustration of the SMB forcing strategies. Under sustained atmospheric warming, the ice sheet retreats from its initial geometry (blue line) to a new geometry (red line). Open black circles illustrate the locations of SMB values defined on the initial ice sheet geometry, while filled black circles indicate the locations where SMB is applied on the evolving geometry. The three forcing strategies shown are: (i) applying SMB directly without accounting for elevation differences (solid black line), (ii) parameterizing the SMB–elevation feedback using runoff gradients from surrounding grid cells (grey lines), and (iii) remapping SMB from grid points at the same elevation onto the current ice sheet, thereby translating the SMB field inland (dashed black line).

The fundamental principle of this method is that when SMB is remapped back to the original ice sheet geometry, the resulting SMB field remains as close as possible to the original forcing data (Fig. 4). This ensures that the SMB forcing retains

maximum consistency with the climate model output while still dynamically adapting to ice sheet geometry changes.

**Figure 2.** Local basins and weighting function applied during remapping. Values increase linearly from 0.0 at the basin boundaries to 1.0 at the center, within a defined transition zone. Weighting values in adjacent basins decrease accordingly. The black contour outlines the present day ice sheet margin.

Figure 3. SMB lookup table per basin for NorESM2-MM-SSP5-8.5 forcing until the year 2100. Coloured lines are given every 5 years.

# 270 2.3.4 SMB anomaly remapping

When using the remapping approach for aSMB, it is not sufficient to remap only the anomalies themselves. To fully account for the effect of surface elevation changes on SMB, the vertical gradient of SMB must also be remapped to the reference (non-evolving) geometry. This is necessary because aSMB is defined as the difference between SMB at a given time and a reference SMB, both evaluated on a fixed topography, which is typically that of the RCM:

$$aSMB(t) = SMB(t) - SMB_ref$$
(4)

When remapping aSMB to the evolving ice sheet surface without further correction, the anomaly implicitly reflects only the time-dependent climate signal but neglects the implicit time dependence introduced by changes in surface elevation. To correctly apply aSMB in a remapping framework however, an elevation correction term based on the vertical SMB gradient has to be included. In addition to remapping the time-dependent SMB anomaly (aSMB(t,h)) to the evolving geometry, the initial SMB gradient (d(SMB\_ref)/dz) has to be remapped to the initial geometry. The full anomaly applied to the evolving surface, ASMB(t,h), is thus given by:

$$ASMB(t,h) \approx R(aSMB(t),h) + R\left(\frac{dSMB\_ref}{dz}, h_0\right) \cdot \Delta h(t), \tag{5}$$

where  $h_0$  is the initial surface elevation,  $\Delta h(t) = h(t) - h_0$ , and  $R(\cdot, h)$  denotes the remapping operator that interpolates fields onto the evolving geometry at time t.

This correction ensures that the applied anomaly reflects not just the temporal climate signal but also the local SMB–elevation relationship. Incorporating this term allows the anomaly to approximate the SMB response to both climate change and evolving topography, and prevents systematic underestimation or misrepresentation of melt rates as the ice sheet retreats. Details are described in Goelzer et al. (2020b) Sect. 4.2 and Appendix A.

#### 3 Results



When remapped to the same topography, the SMB remapping procedure effectively preserves the overall structure of the fixed elevation MAR SMB field (Fig. 4 a), maintaining its general spatial patterns. Accumulation and ablation zones closely agree with the original forcing field. However, it introduces a smoothing effect, particularly in regions with steep SMB gradients. This is most pronounced in the South-East, where high SMB values are underestimated compared to the original fields. This effect stems from the interpolation inherent in the remapping process, which averages over localized extremes and smoothes over basin boundaries, reducing sharp contrasts and dampening small-scale variability. Differences of all fields to the fixed elevation MAR SMB field are visualized in Fig. 4 (e) - (g).

While differences between total SMB remapping and aSMB remapping are generally small, localized deviations can be observed in certain regions. For example, in the mid-west outlet glacier region, the remapped total SMB field exhibits stronger gradients compared to the aSMB case. A possible explanation for this difference is that in the total SMB remapping approach, both the mean state and the spatial gradients of the SMB field are adjusted to reflect changes in topography, whereas the aSMB method retains the spatial structure of the original reference SMB. As a result, aSMB remapping may underrepresent spatial variations where strong topographic control influences SMB patterns. In contrast, total SMB remapping can amplify local gradients as the underlying SMB field shifts along with evolving ice geometry.

# 3.1 Centennial time-scale

Figure 5 shows the cumulative mass change simulated with each of the four SMB-forcing strategies. Over centennial timescales, differences between the various SMB forcing methods remain relatively small. By the year 2100, total ice mass loss differs

**Figure 4.** Upper panels: SMB fields at year 2015 for NorESM2-MM–MAR forcing under four forcing modes: (a) no SMB–elevation feedback, (b) parameterized SMB–elevation feedback, (c) remapped SMB anomalies, and (d) remapped total SMB. Lower panels: Differences relative to (a) for the same year and forcing modes. The black contour indicates the ice sheet extent. Note that in panel (e), no visible difference from the original SMB field appears because no changes in ice sheet topography have occurred at this stage.

by 3.4 Gt across the tested methods. This difference is well within the range of structural model and parameter uncertainty and suggests that over the centennial time scale, the specific treatment of SMB-elevation feedback is less critical for mass loss projections.

Among the different approaches, the simulation omitting any SMB-elevation feedback results in the least mass loss, followed by the one using the runoff-gradient-based parameterization. Slightly greater mass loss occurs when using SMB anomaly remapping, while the greatest loss is observed with total SMB remapping. This ordering reflects the increasing degree to which each method captures the dynamic interaction between surface mass balance and evolving ice sheet geometry.

**Figure 5.** Projections forced with regionally downscaled NorESM2-MM forcing under SSP5-8.5 in various forcing modes: SMB anomalies at fixed ice sheet geometry (blue), SMB anomalies with parametrized elevation feedback (green), remapped SMB anomalies (orange), total SMB remapped (pink).

#### 315 3.2 Multi-millennial time-scale




Over the multi-millennial timescale, differences between methods become increasingly evident, as feedbacks between topography and surface climate reinforce. SMB fields and ice sheet geometries evolve differently depending on the SMB forcing method. Over the first few thousand years, it becomes evident that the simulation without elevation feedback fails to produce sufficient ablation to significantly reduce ice volume (Fig. 6). In contrast, the simulation with a parameterized SMB–elevation feedback shows a more pronounced and spatially extensive ablation zone, especially in the South-West, where low-elevation ice becomes increasingly vulnerable to warming. However, once the ice sheet margins retreat towards higher elevations in the interior, the runoff gradient-based parameterization becomes less effective. In these regions, runoff gradients are weak and the parameterization fails to produce a sufficiently wide ablation zone. As a result, retreat slows, and the ice sheet prematurely stabilizes despite continued climate forcing.

SMB and SMB anomaly remapping approaches produce a more extensive ablation zone than the parameterized SMB–elevation feedback, resulting in a more pronounced and continuous decline of the ice sheet. These differences become apparent after approximately 2000 years of simulation. The spatial structure of the ablation zone in the remapped methods is smoother but more extensive, although total SMB remapping produces sharper SMB gradients near retreating margins.

These distinctions between the methods lead to pronounced differences in the long-term evolution of ice sheet mass (Fig. 7). After approximately 2,000 years, simulations using the parameterized SMB-elevation feedback reach a quasi-stable state, with the ice sheet stabilizing at around  $1.6 \times 10^6$  Gt, while simulations using total SMB or SMB anomaly remapping continue

to lose mass. The larger and more persistent ablation zones produced by the remapping methods drive continued retreat, preventing the ice sheet from approaching a new equilibrium, even after 10,000 years of simulation. The simulation without any SMB–elevation feedback results in significantly lower ice mass loss compared to all other approaches, which demonstrates that that some explicit representation of this feedback is essential in long-term simulations.

Figure 6. SMB fields for simulations over 10.000 years using NorESM2-SSP5-8.5 forcing in different forcing modes.

**Figure 7.** Projections forced with regionally downscaled NorESM2-MM-forcing (extended) under SSP5-8.5 and for various forcing modes: SMB anomalies at fixed ice sheet geometry (blue), SMB anomalies with parametrized elevation feedback (green), remapped SMB anomalies (orange), total SMB remapped (pink), total SMB remapped with glacial-isostatic adjustment (dashed pink).

So far we have focussed our analysis on comparing the different SMB forcing approaches without accounting for the effect of glacial-isostatic adjustment. We now include the isostatic adjustment in a simulation using the SMB remapping method and compare the results in Fig.8. Over the multi-millennial timescale, the influence of isostatic rebound becomes increasingly important. After 10.000 years the rebound run maintains about 25 % more ice mass than the no-rebound run. This reduction in ice loss results from the gradual uplift of the bedrock in response to ice unloading, which alters the surface elevation and thereby affects the applied SMB when elevation feedback is considered. The resulting higher elevations reduce melt rates and contribute to a delayed retreat of the ice sheet.

**Figure 8.** Ice sheet evolution under NorESM2-MM-SSP5.8-5 remapped SMB forcing: neglecting glacial isostatic adjustment (upper panels), including glacial isostatic adjustment (lower panels).

## 345 3.3 Sensitivity to ESM and SSP uncertainty


Using the novel remapping approach, together with glacial-isostatic adjustment, we now further investigate the future evolution and long-term stability of the Greenland ice sheet under a broad suite of climate forcings. The experiments are forced with MAR, which is driven at its boundaries by output from six CMIP6 ESM–SSP combinations: UKESM1-0-LL-SSP5-8.5, NorESM2-MM-SSP5-8.5, MPI-ESM1-2-HR-SSP5-8.5, NorESM2-MM-SSP2-4.5, CESM2-SSP1-2.6, and MPI-ESM1-2-HR-SSP1-2.6. Together these cases span the upper, middle, and lower ends of the CMIP6 projection range as highlighted by Rahlves et al. (2025) (their Fig. 8), giving us a representative spread of possible twenty-first-century climates and their millennial extensions. Extensions beyond 2100 are constructed by repeating the 2081–2100 climatology from each ESM (see Sect. 2.3).

Differences in the simulated ice sheet response emerge within decades and become increasingly pronounced over the millennial time scale. Under most climate forcing scenarios, the ice sheet stabilizes after approximately 1.000 to 4.000 years of simulated time, though the final ice volume varies substantially between scenarios (Figs. 9 and 10). For instance, under the high-emissions and high climate sensitivity UKESM1-0-LL-SSP5-8.5 scenario, the ice sheet undergoes complete disintegration shortly after 4,000 years. In contrast, under low-emissions scenarios such as CESM2-SSP1-2.6 and MPI-ESM1.2-SSP1-2.6,

the ice sheet retains the majority of its mass, stabilizing at a reduced state.


Notably, the NorESM2-MM-SSP5-8.5 and MPI-ESM1.2-SSP5-8.5 scenarios do not lead to a new equilibrium within the 10,000-year simulation period. In these cases, the ice sheet continues to exhibit a slightly negative mass balance even after losing more than half of its initial mass, suggesting the potential for ongoing long-term retreat without stabilization under sustained high-emissions forcing.

**Figure 9.** Projections forced with remapped SMB forcing for various ESM-scenarios (all including glacial isostatic adjustment). For simplicity, the MAR SMB fields and corresponding CISM simulations are referred to by the name of their forcing ESM and scenario.

**Figure 10.** Ice sheet after 10,000 years of simulation under various climate forcing scenarios. (a) UKESM1-0-LL-SSP5-8.5, (b) NorESM2-MM-SSP5-8.5, (c) MPI-ESM1-2-HR-SSP5-8.5, (d) NorESM2-MM-SSP2-4.5, (e) CESM2-SSP1-2.6, (f) MPI-ESM1-2-HR-SSP1-2.6.

## 365 4 Disscussion







Our results confirm the importance of accounting for evolving ice sheet geometry when applying SMB forcing in long-term simulations of the Greenland ice sheet, particularly under sustained high forcing scenarios. While differences between SMB forcing methods are modest over the first few centuries, simulated ice volume begin to diverge significantly over multimillennial timescales (Fig. 7). Simulations that neglect the SMB-elevation feedback significantly underestimate long-term mass loss and fail to reproduce realistic retreat patterns. The standard runoff-gradient parameterization partially mitigates this bias. However, this approach remains limited as soon as the ice sheet margins retreat into regions with weak runoff gradients, such as the interior of the ice sheet. This method underestimates the inland expansion of the ablation zone as the ice sheet retreats. In contrast, the SMB and aSMB remapping approaches, presented here, dynamically adjust SMB fields according to evolving geometry, which leads to a more spatially consistent representation of ablation processes and supports a more realistic projection of long-term ice sheet retreat.

It should be noted that the MAR and CISM topographies are not identical. However, the mismatch is limited because MAR is run on observed topography and CISM is initialized to closely match the observed state. While this small mismatch can introduce local inconsistencies in the applied forcing, particularly in steeply sloping regions, it is identical across all forcing strategies and therefore does not affect the differences between them.

At the start of the experiments the SMB fields produced by aSMB remapping share similar details with those produced with the parameterized SMB-elevation feedback, such as, for example, distinct positive SMB in the mid-west outlet-glacier region. This is expected, because both techniques superimpose the same fixed reference SMB onto their respective anomaly terms, thereby inheriting identical patterns of orographic precipitation and coastal ablation. Under the NorESM2-MM-SSP5-8.5-scenario, the similarity persists for about 4000 years, during which most of the mass loss is still confined to the margin. Beyond that point the two methods begin to diverge. The standard parameterization scheme relies on runoff gradients that become very weak in the interior; as a result it struggles to translate margin retreat into sufficiently negative SMB further inland and the ablation zone stalls. In contrast, aSMB remapping moves the full anomaly field along with the changing topography, letting the band of negative SMB migrate inward.

Differences between remapped SMB anomalies and total SMB remapping are relatively small in terms of evolution of the ablation zone and ice volume. However, total SMB remapping produces steeper negative gradients at the retreating margins and therefore produces slightly more negative SMB in the ablation zone. This is because total SMB remapping implicitly updates both the anomaly and the reference SMB together, thereby preserving the steep elevation-dependent gradients even as the ice sheet topography changes. As warming continues, the SMB anomalies grow in amplitude relative to the static reference field, which reduces the influence of whether the reference is remapped or held fixed. Therefore, remapping the underlying reference SMB field together with the anomalies or leaving it unchanged has only a negligible effect, and both methods produce very similar mass-loss outcomes at large scales.

At the regional to local scale, however, the two methods show some differences. Total SMB remapping smooths the entire SMB field, which is an inherent effect of the interpolation. In the aSMB remapped field, only the anomaly is smoothed, while

the reference SMB remains unchanged. As a result, aSMB remapping preserves features from the reference SMB, for example, the high-precipitation signature of the south-east mountains. Ultimately, the choice between the two approaches represents a trade-off: while it has only a second-order effect on long-term ice-volume change, it influences the sharpness of SMB gradients near the margin and the persistence of orographic accumulation peaks, with the magnitude of these effects depending on the strength of the applied forcing.






An important practical advantage of the SMB remapping approach compared to other approaches to simulate the long-term evolution of the Greenland ice sheet lies in its computational efficiency. It enables extended ice sheet simulations without the need for fully coupled climate—ice sheet models, which are often too expensive for multi-millennial or ensemble studies (Bind-schadler et al., 2013; Goelzer et al., 2020a). The remapping method allows for extensive scenario testing, which is critical for assessing the stability of the Greenland ice sheet under a wide range of future climate trajectories.

We also find that a simplified extension of SMB forcing, such as using the 20-year average beyond 2100, does not significantly alter results when compared to a repeated shuffling of the yearly forcing over the last 20 years. This is consistent with previous studies indicating that the omission of interannual variability in SMB forcing has only a minor impact on long-term simulation outcomes (e.g. Lauritzen et al., 2023; Zolles and Born, 2024; Verjans et al., 2025). This simplification further reduces the complexity of the modeling workflow and supports the method's suitability for extended simulations spanning multiple millennia.

Although the main goal of this study is methodological, our test ensemble still provides a tentative glimpse of long-term outcomes. The runs suggest that any eventual ice sheet stabilization is highly sensitive to both the emissions pathway and the choice of ESM. Our results indicate that under low and medium emission scenarios, the ice sheet may be stabilized after 2.000 to 4.000 years at a reduced ice mass of ca. 85 - 95 % of its present-day mass. In contrast, all high-emission scenarios result in substantial mass loss. For example, the UKESM-SSP5-8.5-forced simulation leads to complete deglaciation within 4,000 years, while the NorESM2-LL-SSP5-8.5 and MPI-ESM-SSP5-8.5 simulations indicate ongoing retreat beyond 10,000 years, after having already lost 55% and 47% of their original mass, respectively.

In addition to ESM dependence, previous studies have shown that the choice of RCM used to downscale ESM output into SMB forcing can also strongly influence long-term ice sheet evolution (Glaude et al., 2024; Goelzer et al., 2025a). While our experiments use only one RCM, it is important to acknowledge this additional source of uncertainty.

Another source of uncertainty arises from the fact that the forcing ESMs do not account for evolving ice sheet topography. Large-scale changes in ice sheet geometry may in turn influence atmospheric circulation and precipitation patterns, an effect that is neglected in our setup. While SMB remapping captures the topographic migration of local climate fields, the omission of such large-scale ice sheet–climate feedbacks may lead to underestimation of circulation-driven changes in SMB over multicentennial to millennial timescales (Vizcaíno et al., 2008; Andernach et al., 2025; Feenstra et al., 2025).

While the factors above relate to external forcing, the SMB remapping method has its own limitations that should also be considered when interpreting results. A key assumption of the method is that present-day relationships between SMB and elevation per region remain valid as the ice sheet evolves. However, this assumption may break down as ice sheet geometry significantly changes, especially near the margins where local climate conditions can diverge from present-day patterns. In

such areas, the similarity that remapping relies on may no longer hold, particularly where local processes like topographically driven wind regimes dominate (Delhasse et al., 2024). The SMB-elevation relationship may also not be preserved in climates that differ fundamentally from the present, such as under altered insolation patterns that drive shifts in atmospheric circulation. The method also assumes a static basin structure, which may become less meaningful as the ice sheet retreats and reorganizes its surface hydrology and ice flow.

Finally, our experiments focus on surface forcing and do not explicitly include evolving ocean conditions beyond the year 2100. As the atmosphere warms, changes in ocean circulation and temperature can affect submarine melt and outlet-glacier retreat. However, because ocean forcing is only relevant while the ice sheet margin remains marine-terminating, its influence on our long-term high-forcing experiments is limited.

Nonetheless, the approach reflects key aspects of the physical processes that govern ice sheet–climate interactions. As the ice sheet retreats, the remapping method effectively shifts climate fields inland, mimicking orographically driven inland migration of precipitation patterns (e.g. Delhasse et al., 2024; Merz et al., 2014b). Similar processes may also apply to other components of the surface energy balance, such as heat fluxes (e.g. Merz et al., 2014a), suggesting that remapped SMB fields may reflect broad-scale climatic evolution beyond just elevation feedbacks.

However, the method does not account for regional feedbacks such as changes in cloud cover, surface albedo, or katabatic wind systems, all of which can significantly influence surface energy balance and SMB (Box et al., 2012; Hofer et al., 2017). These feedbacks are implicitly neglected in stand-alone ice sheet models, and their exclusion represents an important caveat when interpreting long-term simulations.

Over multi-millennial timescales, uncertainty inevitably increases due to evolving boundary conditions and unresolved feedbacks between climate and the ice sheet. Still, simulations on these timescales are valuable for identifying potential stability thresholds and exploring a large range of future ice sheet trajectories.

Conceptually, the remapping approach shares some similarities with elevation class methods (Sellevold et al., 2019; Petrini et al., 2025). While elevation class methods are designed primarily to compensate for model resolution gaps, they also provide a way to adapt to dynamic changes in topography. In our case, the remapping approach allows for climate forcing to follow topography laterally, which effectively captures processes such as the migration of anticyclones along retreating slopes, something that elevation class methods do not. This difference underlines the greater spatial adaptability of the proposed remapping process and its utility for modeling large-scale, long-term ice sheet change.

Finally, SMB remapping enables the use of high-resolution, physically-based forcing in stand-alone models, offering a significant improvement over simplified parameterizations such as PDD models. This allows for more realistic and adaptable long-term ice sheet simulations, particularly as long as fully coupled models are constrained by computational cost.

# 5 Summary and Conclusions



We propose SMB remapping for stand-alone ice sheet modeling as a computationally efficient method to simulate the longterm evolution of the Greenland ice sheet. This method allows the use of high-resolution RCM output and dynamically adapts SMB fields in response to changes in ice sheet geometry, effectively capturing first-order interactions between climate forcing and ice sheet evolution.

A key advantage of the SMB remapping technique is its ability to preserve the spatial structure of the ablation zone, even as the ice sheet margins substantially retreat. This reduces non-physical biases and provides more realistic projections of ice sheet evolution, especially over multi-millennial timescales when large geometric changes are expected. SMB remapping therefore provides a crucial improvement over methods that apply static SMB fields or rely on simplified SMB–elevation parameterizations, which tend to underestimate long-term mass loss, particularly in the ice sheet interior where runoff gradients are weak.

In summary, SMB remapping represents an efficient technique for incorporating elevation-driven SMB feedback into standalone ice sheet models. It allows for the application of high-quality forcing, while keeping more realistic ablation zone structure during the long-term evolution of the ice sheet. The method is particularly well-suited for long-term and ensemble simulations, offering a valuable alternative to fully coupled climate—ice sheet models, which are often limited by their high computational cost. Although limitations remain, especially in representing feedbacks beyond elevation, the method provides an effective tool for exploring future ice sheet trajectories across a wide range of climate scenarios.

Code and data availability. The CISM code version used in this study is available at https://doi.org/10.5281/zenodo.17507385 (Rahlves and Goelzer, 2025). Simulation output and forcing used in this study, as well as scripts to produce remapped SMB forcing are available via the Sigma2 Research Data Archive (NIRD RDA) at https://doi.org/10.11582/2025.i0svqwbs (Rahlves, 2025).

*Author contributions.* HG designed the study and supervised the work. CR performed all experiments and analyzed the results. The manuscript was written by CR with input and critical feedback from all authors.

Competing interests. The authors declare that they have no conflict of interest.

*Financial support.* CR, HG and PML have received funding from the Research Council of Norway under project 324639, GREASE. HG has received funding from the European Union's Horizon 2020 research and innovation programme under grant agreement 869304 (PROTECT).

Acknowledgements. HG acknowledges fruitful discussions with Roderik van de Wal, Michiel van den Broeke, and Brice Noël at early stages of this work. Furthermore, we acknowledge Xavier Fettweis and the MAR group for providing regionally downscaled climate forcing as well as the World Climate Research Programme (WCRP) and its Working Group on Coupled Modelling for coordinating and promoting CMIP6. We thank the climate modeling groups for producing and making available their model output, and the Earth System Grid Federation (ESGF)

for archiving the CMIP data and providing access. Resources for this work were provided by Sigma2 - the National Infrastructure for High Performance Computing and Data Storage in Norway through projects NN8085K, NN8006K, NS5011K, NS8006 K and NS8085K.

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
