# Peer review of "Investigating the multi-millennial evolution and stability of the Greenland ice sheet using remapped surface mass balance forcing"

_EGUsphere, 2025_

## Referee Comment (RC1)

**A review of « Investigating the multi-millennial evolution and stability of the Greenland ice sheet using remapped surface mass balance forcing », Rahlves et al., 2025.**

Rahlves and co-authors present a new method to account for the melt–elevation feedback without requiring full coupling between atmospheric and ice sheet models over Greenland. This alternative to computationally expensive coupling directly modifies the SMB forcing fields within CISM to adapt them to year-by-year changes in surface elevation. This method, referred to as SMB remapping, is evaluated against other approaches commonly used in the literature over both short (centennial) and long (multi-millennial) time-scales. After comparing its performance with existing methods, the authors include the representation of glacial isostatic adjustment in simulations using the approach they consider most robust for capturing melt–elevation feedback. They also compare projections driven by SMB fields derived from various emission scenarios across different ESMs. In conclusion, SMB remapping proves to be a valuable approach for accounting for melt–elevation feedback, as it reduces biases and uncertainties associated with conventional parameterizations and better represents the structure of the ablation zone in projections.

This method is original and appears promising. It will likely be employed in future studies. The manuscript is well written and well structured. I recommend accepting the paper with only minor revisions, as outlined in the comments below.

**Main comments**

- From a general perspective, I recommend providing more detail in the Experimental setup section. Although the authors reference other papers for different part of the method, the originality of this study relies on the methodology itself, so it is essential to describe it as clearly as possible. See the minor comments, but feel free to go beyond them to improve clarity. For instance, I also suggest including a figure to illustrate the 4 methods used for considering the feedback. This section could also be made more consistent in the description of the different experiments, for example, by standardizing the variable names in the equations (see minor comment).
- The discussion is already well developed, but I recommend addressing these few following points:
    - Discuss the influence of the differences in initialed topography for CISM and MAR topography (on which the SMB is computed).
    - NorESM and other ESMs used do not account for evolving ice sheet topography. This could be a source of uncertainty as changes in topography of the ice sheet may influence large-scale climate circulation. What would be the potential influence on your results?

o   On shorter time-scales, depending on the ocean conditions considered here (see minor comment), are there any uncertainties to mention? As the atmosphere warms and changes, ocean conditions also evolve. This may affect the mass loss of the ice sheet, and I guess this could add some uncertainties to the results obtained here, as far as ice sheet is not completely retreated inland.

***Minor comments***

L40: "Simplified physics" for atmosphere ocean and land/polar surface processes?

L120: How CISM considers icebergs and their contribution to the total mass balance?

L120: Could you precise what CISM is considering for ocean conditions?

L142: MAR v3.12 (Fettweis et al., 2017) → more recent reference actually using MARv3.12: Lambin et al. (2023)

L142: As I guess MAR didn't run on CISM 4km grid, did you receive the SMB and ST products from MAR already interpolated on the CISM 4km grid? If not, could you precise how did you interpolated it on the 4km grid? And I guess that this interpolation is the first step before using it to calculate your SMB anomalies and remapped SMB? Precise if necessary.

L145-146: "Outlet-glacier retreat is prescribed via retreat masks up to 2100, after which the mask is held fixed." Does it mean that the ice sheet is retreating with a constant rate after 2100? Please clarify here.

L141-147: When saying that the mean value is repeated, I guess you're talking about the SMB values used as forcing for the ice sheet model. Please clarify.

L160: Could you also specify to what you refer with SMB_ref_ERA5 ? I guess it's the annual mean SMB for the reference period (1960-1989) from MAR-downscaled ERA5 SMB.

L163: An extra figure illustrating all your 4 methods could be interesting to well understand how these 4 methods are working, and what's common or different between them.

L168: If I understand well, SMB(h_fixed) = SMB_ESM(t) from equation (2). If these 2 variables are referring to the same thing, could you rename with the same name? This way, it could be easier to compare methods.

L176: I would add "total" or "full-SMB" (+ and anomalies remapping) in this title to be clearer and not be confused with the title of point 2.3.4. Or call it remapping method.

L222: As you used the mean SMB 2180-2100 to extend your simulations, I guess you also used a same lookup table from 2100 to the end of your simulations? If yes, could you

precise it in the text as well as if it's a "mean lookup table" of 2180-2100, or the one in 2100,...? Otherwise, could you detail what's used after 2100?

L229-231: As I'm not sure to well understand how exactly you interpolate the SMB values from the lookup table with the new elevation of the model (and the basin classification), could you be a bit more specific for the points 2 and 3?

Figure 4: It could be useful to display the SMB differences here instead of in the Appendix. Differences are more visible. You could perhaps merge both Figure 4 and A1 into one and refer to this one in your Appendix. Because you're describing these differences in an entire paragraph (L268-274).

L278: I guess you didn't remove any drift of your model of these results. But, if you have quantified it, could you mention it and compare it to the differences you obtained here (3.4Gt) when explaining that this value is smaller than the uncertainty of your model, or detail this uncertainty?

L375-376: "The runs suggest that any eventual ice sheet stabilization is highly sensitive to both the emissions pathway and the choice of ESM." I suggest also to add, here, or in another paragraph talking about the RCM, that it's also dependent of the RCM used to downscale ESM's climate and "translate" it into SMB.

***Typo***

L54: "extending to the year 2300" → 2200.

L67: "of of the ice sheet" → of the ice sheet

L129: (Broeke et al., 2009) → (Van den Broeke et al., 2009), same in the reference list.

L138: (see Fig. 6 in Rahlves et al. (2025)) → (see Fig. 6 in Rahlves et al., 2025).

L411 : «adaptability of of » → adaptability of

***References***

Lambin, C., Fettweis, X., Kittel, C., Fonder, M., & Ernst, D. (2023). Assessment of future wind speed and wind power changes over South Greenland using the Modèle Atmosphérique Régional regional climate model. International Journal of Climatology, 43(1), 558–574. https://doi.org/10.1002/joc.7795

---

## Referee Comment (RC2)

**Review of egusphere-2025-2192: Investigating the multi-millennial evolution and stability of the Greenland ice sheet using remapped surface mass balance forcing**

Rahlves et al. adapt an SMB remapping method developed by Goelzer et al. (2020) in order to parameterize the effect of the melt/SMB-elevation feedback in long term standalone Greenland ice sheet simulations.
Fully coupled climate and ice sheet simulations are not possible on long time scales and uncoupled ones don't take into account the effect of the melt/elevation feedback due to the use of a fixed ice geometry. As confirmed by Rahlves et al. in this study, while not taking this feedback into account doesn't matter much over the next century, ignoring the increased melt resulting from it leads to important differences in projected mass loss on millennial time scales. Surface mass balance is the link between the atmosphere and the ice sheet and finding an efficient way to parameterize the effect of the melt/elevation feedback on the SMB when it is used as forcing in an ice sheet model simulation is therefore crucial for the study of the long term stability of the GrIS. This new computationally efficient way of including the melt/elevation feedback in standalone ice sheet simulations proposed by Rahlves et al. is therefore extremely relevant to cryospheric science.

The paper is generally very well written and justified throughout. In particular, I really liked the thorough summary of available methods and their shortcomings and advantages. I only have relatively minor comments and recommend that this paper is accepted, provided that the authors address the following comments.

**1. Minor comments**

**1.1 General comment on reanalysis/ESM-forced MAR SMB and CISM**

The SMB you're using in this study is an output of MAR, whose boundaries have been forced by a range of ESMs and reanalysis, but it's sometimes a bit vague in the text after the initial mention. Even if, here, the models are not the primary focus of the paper, it's easy for the reader to forget the SMB doesn't directly come from the reanalysis or ESMs (especially since at least UKESM and CESM can compute their own SMB) without a reminder here and there in the text.

In particular, I would modify the following sentences and sections:

- p5, line 137: **The historical run is forced with MAR-dowscaled ERA5 SMB and ST.** Written this way it makes it sound like MAR directly downscales the SMB calculated in ERA5 whereas it downscales fields like temperature and humidity and calculates SMB in its own surface/snow module. Can you rewrite the sentence to make it less ambiguous?

- p6, section 2.3.1: various things in this sections are a bit confusing.
  First, I would change the first sentence a bit (see below).
  Then, below eq. (1) I would change SMB_ref_ERA5 to SMB_ref to keep the description of more general and not attach it to a specific reanalysis/model.

Also, anomaly-based SMB methods are pretty common to e.g. address problems linked to possibly large biases in ESMs but it would still be nice to have a mention of why.

→ You could modify the text with something along the lines of
**As a baseline experiment, we compute the SMB at time step t as the reference SMB (here ERA5-forced MAR SMB over 1960-1989) to which we add anomalies of the respective ESM SMB with respect to its mean over the reference period (1960-1989):**

**SMB(t) = SMB_ref + SMB_anomaly(t)                              (1)**

**where SMB_ref is the reference SMB and SMB_anomaly(t) is the ESM anomaly at time step t, i.e.**

**SMB_anomaly(t) = SMB_ESM(t)-SMB_ref_ESM            (2)**

**In this approach, … not accounted for. As is often the case (refs), we also use anomalies with respect to a reference SMB field because … .**

- p14, section 3.3 Sensitivity to ESM and SSP: After the list of forcing ESMs and scenarios, you could mention that the MAR SMB and CISM forced simulations are later referred to by the name of the forcing ESM and scenario to remind the reader one last time that the SMB is computed in the ESMs themselves.

Finally, as some of the forcing ESMs also work as fully coupled climate and ice sheet models, I would mention CISM here and there as well to further remind the reader that, when they read e.g. UKESM1-0-LL-SSP-8.5 in the legend of figure 9, the forcing ESM is just the first step in a "3-part simulation", i.e. the UKESM climate forced MAR boundaries, which computes the SMB that is remapped and finally used as forcing in CISM.

**1.2 Specific comments**

- p2, lines 25-29: here you mention that not taking into account the melt-elevation feedback leads to large biases in mass loss over large timescales but you only mention that SMB from RCMs is mostly computed on a fixed geometry much later in your review of methods. If possible, I would move it forward to this part of the introduction — if you can manage to do that without disrupting the flow later in the introduction.

- p2, lines 44-51: Sellevold et al. (2019) and Petrini et al. (2025) both use an elevation class downscaling method but in a 1-way coupling where the ice sheet geometry  changes aren't known by the atmosphere and land surface (either because the ISM isn't communicating back to the atmosphere in the case of Sellevold or because the outputs of the ESM force a standalone ISM simulation in Petrini). As shown by Feenstra et al. (2025) in their comparison between a 1-way and 2-way coupled CESM-CISM simulation, this can lead to biases in

simulated SMB and mass loss. Since the elevation class method is also commonly used in fully coupled ESM-ISM like UKESM-ice (Smith et al., 2021) and CESM-CISM (Feenstra et al. 2025) and, as you already mention fully coupled ESM-ISM earlier in the introduction, it would be worth mentioning this distinction.

**Feenstra et al, 2025: Role of elevation feedbacks and ice sheet–climate interactions on future Greenland ice sheet melt, https://doi.org/10.5194/tc-19-2289-2025**

- p5, line 148: **Beyond 2100, the forcing is extended by averaging the final 20 years (2080–2100) and repeating this mean value at annual time steps. We verify that shuffling the sequence within this window does not significantly affect the results.**
  I only understood that you meant that it doesn't really matter wether you use a 20-year average of SMB or if you use SMB from individual years randomly shuffled within that time period in the discussion (when you write **compared to a repeated shuffling of the yearly forcing**). Could you rewrite the second sentence to make it more clear?

- p6, eq 3 + L169: use $\Delta h$ instead of dh as you did in equation 5. If I remember my calculus classes correctly, d or $\partial$ are used for rates (as in dRU/dz) whereas differences/ranges should be written as $\Delta$.

- p8, line 222: Figure 3 is referred to in the text before figure 2. I'd put a reference to figure 2 earlier in the text (when you first mention dividing the ice sheet into 25 basins or in step 1.1) so figures are in the order they're referred to.

- p11, line 263: does **original forcing field** refer to the fixed elevation SMB anomaly of NORESM-forced MAR SMB with respect to the ERA-forced reference SMB (from section 2.3.1)? In any case, can you refer directly to Fig. 4a there to make the read easier?

- p12, line 301: isn't the parameterized SMB-elevation feedback simulation the one with a final volume of around $1.6 \times 10^{18}$ Gt (green line) and the 2.4 one the fixed geometry one (blue line)? Also, it should be $10^6$ Gt according to the figures and not $10^{18}$.

**2. Figures**

Most of the figures (apart from 1 and 2) are quite narrow and would benefit from taking the whole width of the page. Figures 3 and 6, in particular, have many panels and it's difficult to see the details mentioned in the text without zooming in a lot.

**3. Typos and grammar**

- p2, L59: standalone. You use the hyphenated stand-alone throughout the manuscript except for this one

- p3, L66 + p18, L 413: physically-based instead of physically based for consistency. Later you use hyphenated versions, e.g. temperature-based, gradient-based, similarity-based …

- p3, L67: remove one of in 'at the surface of of the ice sheet'

- p3, L86: extra space in 'consistent/ uniform'

- p5, L129: van den Broeke et al. instead of Brooke et al. + p20 for the reference

- p5, L132: Pollard and DeConto (Capital D and C and no space between De and Conto) instead of Pollard and Conto + p23 for the reference

- p5, L141 + p14, L318: downscaled instead of down-scaled as you use downscaled most of the time

- p6, l163: calculating the SMB forcing (missing r)

- p16, L339: remove comma between parameterization and partially

- p17, L368: there might be an extra space at the beginning of this sentence

---

## Author Comment (AC1)

**Author's reply to referee comments**

We thank the referee for their thoughtful and constructive comments, which will help improve the clarity and precision of our manuscript. Below we provide point-by-point responses.

**Reply to comments by Referee 2 (RC2):**

**Response to minor comments:**

**Comment: 1.1 General comment on reanalysis/ESM-forced MAR SMB and CISM**
The SMB you're using in this study is an output of MAR, whose boundaries have been forced by a range of ESMs and reanalysis, but it's sometimes a bit vague in the text after the initial mention. Even if, here, the models are not the primary focus of the paper, it's easy for the reader to forget the SMB doesn't directly come from the reanalysis or ESMs (especially since at least UKESM and CESM can compute their own SMB) without a reminder here and there in the text.

**Response:** Agreed. We will consistently clarify that the SMB used is computed by MAR, which is forced by reanalysis/ESM fields, not directly derived from them.

**Comment:** p5, line 137: **The historical run is forced with MAR-dowscaled ERA5 SMB and ST.** Written this way it makes it sound like MAR directly downscales the SMB calculated in ERA5 whereas it downscales fields like temperature and humidity and calculates SMB in its own surface/snow module. Can you rewrite the sentence to make it less ambiguous? You could modify the text with something along the lines of **As a baseline experiment, we compute the SMB at time step t as the reference SMB (here ERA5-forced MAR SMB over 1960-1989) to which we add anomalies of the respective ESM SMB with respect to its mean over the reference period (1960-1989):**

**SMB(t) = SMB_ref + SMB_anomaly(t) (1)**

**where SMB_ref is the reference SMB and SMB_anomaly(t) is the ESM anomaly at time step t, i.e.**

**SMB_anomaly(t) = SMB_ESM(t)-SMB_ref_ESM (2)**

**In this approach, ... not accounted for. As is often the case (refs), we also use anomalies with respect to a reference SMB field because ... .**

**Response:** Thank you for the suggestions, we will revise the sentence as suggested to clarify that MAR computes the SMB using downscaled meteorological inputs (e.g., temperature, humidity) from ERA5, not precomputed SMB.

**Comment:** p6, section 2.3.1: various things in this sections are a bit confusing. First, I would change the first sentence a bit (see below). Then, below eq. (1) I would change SMB_ref_ERA5 to SMB_ref to keep the description of more general and not attach it to a specific reanalysis/model. Also, anomaly-based SMB methods are pretty common to e.g. address problems linked to possibly large biases in ESMs but it would still be nice to have a mention of why.

**Response**: We will rephrase for clarity, use more general notation, and include a brief rationale for using anomaly-based methods, citing relevant references.

**Comment:** p14, section 3.3 Sensitivity to ESM and SSP: After the list of forcing ESMs and scenarios, you could mention that the MAR SMB and CISM forced simulations are later referred to by the name of the forcing ESM and scenario to remind the reader one last time that the SMB is computed in the ESMs themselves.

**Response:** Agreed. We will add a clarifying sentence to remind the reader of this naming convention.

**Comment:** Finally, as some of the forcing ESMs also work as fully coupled climate and ice sheet models, I would mention CISM here and there as well to further remind the reader that, when they read e.g. UKESM1-0-LL-SSP-8.5 in the legend of figure 9, the forcing ESM is just the first step in a "3-part simulation", i.e. the UKESM climate forced MAR boundaries, which computes the SMB that is remapped and finally used as forcing in CISM.

**Response:** We will clarify this workflow in the revised manuscript.

**Response to specific comments:**

**Comment:** p2, lines 25-29: here you mention that not taking into account the melt-elevation feedback leads to large biases in mass loss over large timescales but you only mention that SMB from RCMs is mostly computed on a fixed geometry much later in your review of methods. If possible, I would move it forward to this part of the introduction — if you can manage to do that without disrupting the flow later in the introduction.

**Response:** We will move this point forward in the introduction to improve the logical flow.

**Comment:** p2, lines 44-51: Sellevold et al. (2019) and Petrini et al. (2025) both use an elevation class downscaling method but in a 1-way coupling where the ice sheet geometry changes aren't known by the atmosphere and land surface (either because the ISM isn't communicating back to the atmosphere in the case of Sellevold or because the outputs of the ESM force a standalone ISM simulation in Petrini). As shown by Feenstra et al. (2025) in their comparison between a 1-way and 2-way coupled CESM-CISM simulation, this can lead to biases in simulated SMB and mass loss. Since the elevation class method is also

commonly used in fully coupled ESM-ISM like UKESM-ice (Smith et al., 2021) and CESM-CISM (Feenstra et al. 2025) and, as you already mention fully coupled ESM-ISM earlier in the introduction, it would be worth mentioning this distinction.
**Feenstra et al, 2025: Role of elevation feedbacks and ice sheet–climate interactions on future Greenland ice sheet melt, https://doi.org/10.5194/tc-19-2289-2025**

**Response:** Thank you, we will incorporate this distinction into the introduction and cite Feenstra et al. (2025) along with other relevant literature.

**Comment:** p5, line 148: **Beyond 2100, the forcing is extended by averaging the final 20 years (2080–2100) and repeating this mean value at annual time steps. We verify that shuffling the sequence within this window does not significantly affect the results.** I only understood that you meant that it doesn't really matter wether you use a 20-year average of SMB or if you use SMB from individual years randomly shuffled within that time period in the discussion (when you write **compared to a repeated shuffling of the yearly forcing**). Could you rewrite the second sentence to make it more clear?

**Response:** Correct. We will reword this sentence to make the meaning and implication more transparent.

**Comment:** p6, eq 3 + L169: use Δh instead of dh as you did in equation 5. If I remember my calculus classes correctly, d or ∂ are used for rates (as in dRU/dz) whereas differences/ranges should be written as Δ.

**Response:** Thank you, we will update the notation accordingly.

**Comment:** p8, line 222: Figure 3 is referred to in the text before figure 2. I'd put a reference to figure 2 earlier in the text (when you first mention dividing the ice sheet into 25 basins or in step 1.1) so figures are in the order they're referred to.

**Response:** Absolutely, we will adjust figure references to maintain logical order.

**Comment:** p11, line 263: does **original forcing field** refer to the fixed elevation SMB anomaly of NORESM-forced MAR SMB with respect to the ERA-forced reference SMB (from section 2.3.1)? In any case, can you refer directly to Fig. 4a there to make the read easier?

**Response:** Yes, this refers to the fixed-elevation MAR SMB anomaly. We will revise the sentence and directly reference Figure 4a.

**Comment:** p12, line 301: isn't the parameterized SMB-elevation feedback simulation the one with a final volume of around $1.6 \times 10^{18}$ Gt (green line) and the 2.4 one the fixed geometry one (blue line)? Also, it should be $10^6$ Gt according to the figures and not $10^{18}$.

**Response:** Yes, both correct. Thank you for spotting this. We will fix the text accordingly.

**Comment: 2. Figures**
Most of the figures (apart from 1 and 2) are quite narrow and would benefit from taking the whole width of the page. Figures 3 and 6, in particular, have many panels and it's difficult to see the details mentioned in the text without zooming in a lot.

**Response:** We will increase the width of Figures 3, 6, and others as appropriate to improve readability.

**Comment regarding Typos and grammar:**

**Response:** Thank you, all identified typos and grammatical issues will be corrected.

---

## Author Comment (AC2)

**Author's reply to referee comments**

We would like to thank the referee for their valuable and constructive comments, which will help improve the clarity and quality of our manuscript. Below we provide point-by-point responses.

**Reply to referee comments 1 (RC1):**

**Main comments:**

**Comment (1):** From a general perspective, I recommend providing more detail in the Experimental setup section. Although the authors reference other papers for different part of the method, the originality of this study relies on the methodology itself, so it is essential to describe it as clearly as possible. See the minor comments, but feel free to go beyond them to improve clarity. For instance, I also suggest including a figure to illustrate the 4 methods used for considering the feedback. This section could also be made more consistent in the description of the different experiments, for example, by standardizing the variable names in the equations (see minor comment).

**Response:** We agree and will expand the *Experimental setup* section, standardize variable names, and include an additional schematic to illustrate the four methods.

**Comment (2):** The discussion is already well developed, but I recommend addressing these few following points:
- Discuss the influence of the differences in initialed topography for CISM and MAR topography (on which the SMB is computed).
- NorESM and other ESMs used do not account for evolving ice sheet topography. This could be a source of uncertainty as changes in topography of the ice sheet may influence large-scale climate circulation. What would be the potential influence on your results?
- On shorter time-scales, depending on the ocean conditions considered here (see minor comment), are there any uncertainties to mention? As the atmosphere warms and changes, ocean conditions also evolve. This may affect the mass loss of the ice sheet, and I guess this could add some uncertainties to the results obtained here, as far as ice sheet is not completely retreated inland.

**Response:** We appreciate these valuable suggestions and will incorporate these points in the discussion of the revised manuscript.

**Response to minor comments:**

**Comment:** L40: "Simplified physics" for atmosphere ocean and land/polar surface processes?

**Response:** We will specify that this refers to simplified representations of atmosphere, ocean, and surface processes.

**Comment:** L120: How CISM considers icebergs and their contribution to the total mass balance?

**Response:** The handling of ice bergs is mentioned in L. 122-123. We will further clarify and expand this description.

**Comment:** L120: Could you precise what CISM is considering for ocean conditions?

**Response:** The handling of ocean conditions are discussed in L.145-146. We will further clarify and expand these descriptions (see also response to comment L145-146).

**Comment:** L142: MAR v3.12 (Fettweis et al., 2017) à more recent reference actually using MARv3.12: Lambin et al. (2023)

**Response:** Thank you, we will use this more recent reference.

**Comment:** L142: As I guess MAR didn't run on CISM 4km grid, did you receive the SMB and ST products from MAR already interpolated on the CISM 4km grid? If not, could you precise how did you interpolated it on the 4km grid? And I guess that this interpolation is the first step before using it to calculate your SMB anomalies and remapped SMB? Precise if necessary.

**Response:** We will include a description of the interpolation procedure used to remap MAR outputs onto the 4 km CISM grid.

**Comment:** L145-146: "Outlet-glacier retreat is prescribed via retreat masks up to 2100, after which the mask is held fixed. " Does it mean that the ice sheet is retreating with a constant rate after 2100? Please clarify here.

**Response** The retreat mask is fixed after 2100, meaning the ice sheet can retreat further but cannot re-advance beyond the mask. We will specify this in the revised manuscript.

**Comment:** L141-147: When saying that the mean value is repeated, I guess you're talking about the SMB values used as forcing for the ice sheet model. Please clarify.

**Response:** Correct, we will clarify that this refers to repeated SMB values used as forcing.

**Comment:** L160: Could you also specify to what you refer with SMB_ref_ERA5 ? I guess it's the annual mean SMB for the reference period (1960-1989) from MAR-downscaled ERA5 SMB.

**Response:** Confirmed, it refers to the annual mean SMB from MAR-downscaled ERA5 (1960–1989). We will clarify this in the revised manuscript.

**Comment:** L163: An extra figure illustrating all your 4 methods could be interesting to well understand how these 4 methods are working, and what's common or different between them.

**Response:** We appreciate the suggestion and agree that a visual summary could help clarify the differences and similarities between the four forcing methods. We are currently exploring how to extend Fig. 1 to include an illustration of all methods and will aim to include this in the revised manuscript if we find a suitable approach.

**Comment:** L168: If I understand well, SMB(h_fixed) = SMB_ESM(t) from equation (2). If these 2 variables are referring to the same thing, could you rename with the same name? This way, it could be easier to compare methods.

**Response:** These are not equivalent. SMB(h_fixed) in Eq. 3 is equal to SMB(t) in Eq. 2. We will unify notation where appropriate and clarify the distinction.

**Comment:** L176: I would add "total" or "full-SMB" (+ and anomalies remapping) in this title to be clearer and not be confused with the title of point 2.3.4. Or call it remapping method.

**Response:** Agreed, we will revise the title to clarify that this refers to full SMB and SMB anomalies.

**Comment:** L222: As you used the mean SMB 2180-2100 to extend your simulations, I guess you also used a same lookup table from 2100 to the end of your simulations? If yes, could you precise it in the text as well as if it's a "mean lookup table" of 2180-2100, or the one in 2100,…? Otherwise, could you detail what's used after 2100?

**Response:** Correct, we use a mean lookup table from 2180–2100. We will state this more clearly.

**Comment:** L229-231: As I'm not sure to well understand how exactly you interpolate the SMB values from the lookup table with the new elevation of the model (and the basin classification), could you be a bit more specific for the points 2 and 3?

**Response:** We will provide a more detailed description of this process.

**Comment:** Figure 4: It could be useful to display the SMB differences here instead of in the Appendix. Differences are more visible. You could perhaps merge both Figure 4 and A1 into one and refer to this one in your Appendix. Because you're describing these differences in an entire paragraph (L268-274).

**Response:** We will merge Figure 4 and A1 to improve clarity and emphasize the SMB differences.

**Comment:** L278: I guess you didn't remove any drift of your model of these results. But, if you have quantified it, could you mention it and compare it to the differences you obtained here (3.4Gt) when explaining that this value is smaller than the uncertainty of your model, or detail this uncertainty?

**Response:** We account for historical "drift" as part of the physical response to past forcing. We will clarify this distinction and expand the discussion of uncertainty.

**Comment:** L375-376: "The runs suggest that any eventual ice sheet stabilization is highly sensitive to both the emissions pathway and the choice of ESM. " I suggest also to add, here, or in another paragraph talking about the RCM, that it's also dependent of the RCM used to downscale ESM's climate and "translate" it into SMB.

**Response:** Agreed, although our study uses only one RCM, we will acknowledge the influence of RCM choice and cite relevant studies.

**Comments referring to Typos:**

**Response:** Thank you, all noted typos will be corrected.

---

## Author Response (AR1)

**Author's reply to referee comments and description of changes**

We would like to thank the referee for their valuable and constructive comments, which helped improve the clarity and quality of our manuscript. Below we provide point-by-point responses.

**Reply to referee comments 1 (RC1):**

**Main comments:**

**Comment (1):** From a general perspective, I recommend providing more detail in the Experimental setup section. Although the authors reference other papers for different part of the method, the originality of this study relies on the methodology itself, so it is essential to describe it as clearly as possible. See the minor comments, but feel free to go beyond them to improve clarity. For instance, I also suggest including a figure to illustrate the 4 methods used for considering the feedback. This section could also be made more consistent in the description of the different experiments, for example, by standardizing the variable names in the equations (see minor comment).

**Response:** We have expanded and clarified the *Experimental setup* section by adding further detail, standardizing variable names in the equations, and improving the consistency of the experiment descriptions. In addition, Figure 1 has been revised to include an illustration of all SMB forcing methods.

**Comment (2):** The discussion is already well developed, but I recommend addressing these few following points:

- Discuss the influence of the differences in initialed topography for CISM and MAR topography (on which the SMB is computed).
- NorESM and other ESMs used do not account for evolving ice sheet topography. This could be a source of uncertainty as changes in topography of the ice sheet may influence large-scale climate circulation. What would be the potential influence on your results?
- On shorter time-scales, depending on the ocean conditions considered here (see minor comment), are there any uncertainties to mention? As the atmosphere warms and changes, ocean conditions also evolve. This may affect the mass loss of the ice sheet, and I guess this could add some uncertainties to the results obtained here, as far as ice sheet is not completely retreated inland.

**Response:** We incorporated these three points in the discussion of the revised manuscript:

- The influence of differences in initialized topography between CISM and MAR (on which the SMB is computed) is now discussed in lines 377ff-380.

- The limitation that NorESM and the other ESMs used do not account for evolving ice sheet topography and the potential implications for large-scale climate circulation is now explicitly addressed in lines 426-430.
- Possible uncertainties related to evolving ocean conditions are discussed in lines 440-443.

**Response to minor comments:**

**Comment:** L40: "Simplified physics" for atmosphere ocean and land/polar surface processes?

**Response:** We specified that this refers to simplified representations of atmosphere, ocean, and surface processes.

**Comment:** L120: How CISM considers icebergs and their contribution to the total mass balance?

**Response:** The handling of ice bergs is mentioned in L. 122-123. We further clarified and expanded this description.

Comment: L120: Could you precise what CISM is considering for ocean conditions?

**Response:** The handling of ocean conditions is discussed in L.145-146. We further clarified and expanded these descriptions (see also response to comment L145-146).

**Comment:** L142: MAR v3.12 (Fettweis et al., 2017) à more recent reference actually using MARv3.12: Lambin et al. (2023)

**Response:** Thank you, we added this more recent reference.

**Comment:** L142: As I guess MAR didn't run on CISM 4km grid, did you receive the SMB and ST products from MAR already interpolated on the CISM 4km grid? If not, could you precise how did you interpolated it on the 4km grid? And I guess that this interpolation is the first step before using it to calculate your SMB anomalies and remapped SMB? Precise if necessary.

**Response:** We added a description of the interpolation procedure used to remap MAR outputs onto the 4 km CISM grid.

**Comment:** L145-146: "Outlet-glacier retreat is prescribed via retreat masks up to 2100, after which the mask is held fixed." Does it mean that the ice sheet is retreating with a constant rate after 2100? Please clarify here.

**Response** The retreat mask is fixed after 2100, meaning the ice sheet can retreat further but cannot re-advance beyond the mask. We specified this in the revised manuscript.

**Comment:** L141-147: When saying that the mean value is repeated, I guess you're talking about the SMB values used as forcing for the ice sheet model. Please clarify.

**Response:** Correct, we rewrote to 'SMB values' to clarify that this refers to repeated SMB values used as forcing.

**Comment:** L160: Could you also specify to what you refer with SMB\_ref\_ERA5? I guess it's the annual mean SMB for the reference period (1960-1989) from MAR-downscaled ERA5 SMB.

**Response:** Confirmed, it refers to the annual mean SMB from MAR-downscaled ERA5 (1960–1989). We rewrote this part and changed the notation to clarify this.

**Comment:** L163: An extra figure illustrating all your 4 methods could be interesting to well understand how these 4 methods are working, and what's common or different between them.

**Response:** We have revised Figure 1 to include an illustration of all SMB forcing methods.

**Comment:** L168: If I understand well, SMB(h\_fixed) = SMB\_ESM(t) from equation (2). If these 2 variables are referring to the same thing, could you rename with the same name? This way, it could be easier to compare methods.

**Response:** These are not equivalent. SMB(h\_fixed) in Eq. 3 is equal to SMB(t) in Eq. 2. We unified notation and clarified the distinction (see also response to comment L. 160).

**Comment:** L176: I would add "total" or "full-SMB" (+ and anomalies remapping) in this title to be clearer and not be confused with the title of point 2.3.4. Or call it remapping method.

**Response:** Agreed, we revised the title to clarify that this refers to full SMB and SMB anomalies.

**Comment:** L222: As you used the mean SMB 2180-2100 to extend your simulations, I guess you also used a same lookup table from 2100 to the end of your simulations? If yes, could you precise it in the text as well as if it's a "mean lookup table" of 2180-2100, or the one in 2100,...? Otherwise, could you detail what's used after 2100?

**Response:** Correct, we use a mean lookup table from 2180–2100. We stated this more clearly.

**Comment:** L229-231: As I'm not sure to well understand how exactly you interpolate the SMB values from the lookup table with the new elevation of the model (and the basin classification), could you be a bit more specific for the points 2 and 3?

**Response:** We have reworded and expanded points 2 and 3 to provide a clearer description of how SMB values are selected from the lookup table based on elevation and basin membership, and how smoothing is applied across basin boundaries to ensure spatial continuity.

**Comment:** Figure 4: It could be useful to display the SMB differences here instead of in the Appendix. Differences are more visible. You could perhaps merge both Figure 4 and A1 into one and refer to this one in your Appendix. Because you're describing these differences in an entire paragraph (L268-274).

**Response:** We have merged Figure 4 and Figure A1 into a single figure to better connect with the main text and to highlight the SMB differences described.

**Comment:** L278: I guess you didn't remove any drift of your model of these results. But, if you have quantified it, could you mention it and compare it to the differences you obtained here (3.4Gt) when explaining that this value is smaller than the uncertainty of your model, or detail this uncertainty?

**Response:** We treat the model's historical "drift" as part of the physical response to past forcing rather than removing it artificially. This is now clarified more explicitly in Sect. 2.2, with a reference to an earlier publication where this issue is addressed in detail. In addition, we have explicitly listed the relevant sources of uncertainty here.

**Comment:** L375-376: "The runs suggest that any eventual ice sheet stabilization is highly sensitive to both the emissions pathway and the choice of ESM." I suggest also to add, here, or in another paragraph talking about the RCM, that it's also dependent of the RCM used to downscale ESM's climate and "translate" it into SMB.

**Response:** Agreed, although our study uses only one RCM, we acknowledged the influence of RCM choice and cited relevant studies.

**Comments referring to Typos:**

**Response:** Thank you, we corrected all noted typos.

**Reply to comments by Referee 2 (RC2):**

**Response to minor comments:**

Comment: 1.1 General comment on reanalysis/ESM-forced MAR SMB and CISM

The SMB you're using in this study is an output of MAR, whose boundaries have been forced by a range of ESMs and reanalysis, but it's sometimes a bit vague in the text after the initial mention. Even if, here, the models are not the primary focus of the paper, it's easy for the reader to forget the SMB doesn't directly come from the reanalysis or ESMs (especially since at least UKESM and CESM can compute their own SMB) without a reminder here and there in the text.

**Response:** Agreed. We clarified that the SMB used is computed by MAR, which is forced by reanalysis/ESM fields, not directly derived from them.

Comment: p5, line 137: The historical run is forced with MAR-dowscaled ERA5 SMB and ST. Written this way it makes it sound like MAR directly downscales the SMB calculated in ERA5 whereas it downscales fields like temperature and humidity and calculates SMB in its own surface/snow module. Can you rewrite the sentence to make it less ambiguous? You could modify the text with something along the lines of As a baseline experiment, we compute the SMB at time step t as the reference SMB (here ERA5-forced MAR SMB over 1960-1989) to which we add anomalies of the respective ESM SMB with respect to its mean over the reference period (1960-1989):

 $SMB(t) = SMB\_ref + SMB\_anomaly(t) (1)$

where SMB\_ref is the reference SMB and SMB\_anomaly(t) is the ESM anomaly at time step t, i.e.

 $SMB_anomaly(t) = SMB_ESM(t)-SMB_ref_ESM(2)$

In this approach, ... not accounted for. As is often the case (refs), we also use anomalies with respect to a reference SMB field because ... .

**Response:** Thank you for the suggestions, we revised the sentence as suggested to clarify that MAR is driven at its boundaries by ERA5 meteorological variables.

**Comment:** p6, section 2.3.1: various things in this sections are a bit confusing. First, I would change the first sentence a bit (see below). Then, below eq. (1) I would change SMB\_ref\_ERA5 to SMB\_ref to keep the description of more general and not attach it to a specific reanalysis/model. Also, anomaly-based SMB methods are pretty common to e.g. address problems linked to possibly large biases in ESMs but it would still be nice to have a mention of why.

**Response**: We rephrased for clarity, improved the notation, and included a brief rationale for using anomaly-based methods, citing relevant references.

**Comment:** p14, section 3.3 Sensitivity to ESM and SSP: After the list of forcing ESMs and scenarios, you could mention that the MAR SMB and CISM forced simulations are later referred to by the name of the forcing ESM and scenario to remind the reader one last time that the SMB is computed in the ESMs themselves.

**Response:** Agreed. We added a clarifying sentence to remind the reader of this naming convention.

**Comment:** Finally, as some of the forcing ESMs also work as fully coupled climate and ice sheet models, I would mention CISM here and there as well to further remind the reader that, when they read e.g. UKESM1-0-LL-SSP-8.5 in the legend of figure 9, the forcing ESM is just the first step in a "3-part simulation", i.e. the UKESM climate forced MAR boundaries, which computes the SMB that is remapped and finally used as forcing in CISM.

**Response:** We clarified this workflow in Sect. 2.

**Response to specific comments:**

**Comment:** p2, lines 25-29: here you mention that not taking into account the melt-elevation feedback leads to large biases in mass loss over large timescales but you only mention that SMB from RCMs is mostly computed on a fixed geometry much later in your review of methods. If possible, I would move it forward to this part of the introduction — if you can manage to do that without disrupting the flow later in the introduction.

**Response:** We decided to leave this section unchanged, as several subsequent paragraphs build on this point, and we believe the overall logic of the introduction flows better this way.

Comment: p2, lines 44-51: Sellevold et al. (2019) and Petrini et al. (2025) both use an elevation class downscaling method but in a 1-way coupling where the ice sheet geometry changes aren't known by the atmosphere and land surface (either because the ISM isn't communicating back to the atmosphere in the case of Sellevold or because the outputs of the ESM force a standalone ISM simulation in Petrini). As shown by Feenstra et al. (2025) in their comparison between a 1-way and 2-way coupled CESM-CISM simulation, this can lead to biases in simulated SMB and mass loss. Since the elevation class method is also commonly used in fully coupled ESM-ISM like UKESM-ice (Smith et al., 2021) and CESM-CISM (Feenstra et al. 2025) and, as you already mention fully coupled ESM-ISM earlier in the introduction, it would be worth mentioning this distinction.

Feenstra et al, 2025: Role of elevation feedbacks and ice sheet-climate interactions on future Greenland ice sheet melt, https://doi.org/10.5194/tc-19-2289-2025

**Response:** Thank you, we rewrote the paragraph to incorporate this distinction and cited Feenstra et al. (2025).

Comment: p5, line 148: Beyond 2100, the forcing is extended by averaging the final 20 years (2080–2100) and repeating this mean value at annual time steps. We verify that shuffling the sequence within this window does not significantly affect the results. I only understood that you meant that it doesn't really matter wether you use a 20-year average of SMB or if you use SMB from individual years randomly shuffled within that time period in the discussion (when you write compared to a repeated shuffling of the yearly forcing). Could you rewrite the second sentence to make it more clear?

**Response:** Correct. We reworded this sentence to make the meaning and implication more transparent.

**Comment:** p6, eq 3 + L169: use  $\Delta h$  instead of dh as you did in equation 5. If I remember my calculus classes correctly, d or  $\partial$  are used for rates (as in dRU/dz) whereas differences/ranges should be written as  $\Delta$ .

**Response:** Thank you, we updated the notation accordingly.

**Comment:** p8, line 222: Figure 3 is referred to in the text before figure 2. I'd put a reference to figure 2 earlier in the text (when you first mention dividing the ice sheet into 25 basins or in step 1.1) so figures are in the order they're referred to.

**Response:** Absolutely, we adjusted figure references accordingly.

**Comment:** p11, line 263: does **original forcing field** refer to the fixed elevation SMB anomaly of NORESM-forced MAR SMB with respect to the ERA-forced reference SMB (from section 2.3.1)? In any case, can you refer directly to Fig. 4a there to make the read easier?

**Response:** Yes, this refers to the fixed-elevation MAR SMB field. We revised the sentence and directly referenced Figure 4a.

**Comment:** p12, line 301: isn't the parameterized SMB-elevation feedback simulation the one with a final volume of around 1.6 x 10^18 Gt (green line) and the 2.4 one the fixed geometry one (blue line)? Also, it should be 10^6 Gt according to the figures and not 10^18.

**Response:** Yes, both correct. Thank you for spotting this. We fixed the text accordingly.

**Comment: 2. Figures**

Most of the figures (apart from 1 and 2) are quite narrow and would benefit from taking the whole width of the page. Figures 3 and 6, in particular, have many panels and it's difficult to see the details mentioned in the text without zooming in a lot.

**Response:** We increased the width of all Figures to improve readability.

**Comment regarding Typos and grammar:**

**Response:** Thank you, we corrected all identified typos and grammatical issues.